# Structure of an Alkaline Pectate Lyase and Rational Engineering with Improved Thermo-Alkaline Stability for Efficient Ramie Degumming

**DOI:** 10.3390/ijms24010538

**Published:** 2022-12-29

**Authors:** Cheng Zhou, Yuting Cao, Yanfen Xue, Weidong Liu, Jiansong Ju, Yanhe Ma

**Affiliations:** 1State Key Laboratory of Microbial Resources, Institute of Microbiology, Chinese Academy of Sciences, Beijing 100101, China; 2College of Life Sciences, Hebei Normal University, Shijiazhuang 050024, China; 3Tianjin Institute of Industrial Biotechnology, Chinese Academy of Sciences, Tianjin 300308, China

**Keywords:** pectate lyase, crystal structure, thermo-alkaline stability, site-directed saturation mutagenesis, ramie degumming

## Abstract

Alkaline pectate lyases have biotechnological applications in plant fiber processing, such as ramie degumming. Previously, we characterized an alkaline pectate lyase from *Bacillus clausii* S10, named BacPelA, which showed potential for enzymatic ramie degumming because of its high cleavage activity toward methylated pectins in alkaline conditions. However, BacPelA displayed poor thermo-alkaline stability. Here, we report the 1.78 Å resolution crystal structure of BacPelA in apo form. The enzyme has the characteristic right-handed β-helix fold of members of the polysaccharide lyase 1 family and shows overall structural similarity to them, but it displays some differences in the details of the secondary structure and Ca^2+^-binding site. On the basis of the structure, 10 sites located in flexible regions and showing high B-factor and positive Δ*T_m_* values were selected for mutation, aiming to improve the thermo-alkaline stability of the enzyme. Following site-directed saturation mutagenesis and screening, mutants A238C, R150G, and R216H showed an increase in the *T*_50_^15^ value at pH 10.0 of 3.0 °C, 6.5 °C, and 7.0 °C, respectively, compared with the wild-type enzyme, interestingly accompanied by a 24.5%, 46.6%, and 61.9% increase in activity. The combined mutant R150G/R216H/A238C showed an 8.5 °C increase in the *T*_50_^15^ value at pH 10.0, and an 86.1% increase in the specific activity at 60 °C, with approximately doubled catalytic efficiency, compared with the wild-type enzyme. Moreover, this mutant retained 86.2% activity after incubation in ramie degumming conditions (4 h, 60 °C, pH 10.0), compared with only 3.4% for wild-type BacPelA. The combined mutant increased the weight loss of ramie fibers in degumming by 30.2% compared with wild-type BacPelA. This work provides a thermo-alkaline stable, highly active pectate lyase with great potential for application in the textile industry, and also illustrates an effective strategy for rational design and improvement of pectate lyases.

## 1. Introduction

Pectin, a complex heteropolysaccharide consisting of α-1,4-linked galacturonate chains, is a naturally ubiquitous constituent of the middle lamella of the primary cell wall in plants [1]. Pectin-degrading enzymes include two main groups: methylesterases that remove methoxyl groups from pectin, and depolymerases (hydrolases and lyases) that cleave the bonds between galacturonate units [2]. Pectate lyases (Pels; EC 4.2.2.2), belonging to the pectin depolymerase group, cleave α-1,4-linked galacturonate units of pectate by β-elimination to generate an unsaturated C4-C5 bond at the non-reducing end of the newly formed oligogalacturonate [3]. Pels are widely distributed in diverse microorganisms and plants, and are major virulence factors in plant pathogens [4]. On the basis of the amino acid sequence, Pels are classified into polysaccharide lyase (PL) families 1, 2, 3, 9, and 10 (http://www.cazy.org/Polysaccharide-Lyases.html accessed on 11 September 2019). Most of the reported Pels belong to the PL1 family, and they generally require additional Ca^2+^ for efficient enzymatic activity [5].

Pels have many potentially important industrial applications, such as in food production and textile processing [6]. Alkaline Pels, which are mostly derived from bacteria and show efficient catalytic activity in alkaline conditions (pH 8–10), have potential for application in industrial processes such as pretreatment of pectic wastewater and degumming of fiber crops in particular [7]. Raw fiber crops usually contain colloidal complex that wraps the fiber and makes the fiber cemented to each other, which means that the fiber cannot be used directly for textiles [8]. So it is necessary to release the fiber from “the bond state” by degumming treatment. Compared with conventional chemical degumming using hot alkaline solutions with or without the application of pressure, enzymatic degumming with alkaline Pels has advantages, including flexible and mild processing conditions, limited damage to fibers, environmentally friendly operation, and easy quality control [1,9]. Because of these advantages and the potential market for enzymatic degumming, many alkaline Pels from microbes with the potential for bioscouring and degumming applications have been reported [9,10,11,12,13,14,15,16]. However, only a few Pels have been found to be suitable for industrial application because of their low stability and activity in the processing conditions (40–70 °C, pH 8–11) [2,6,17]. The discovery of alkaline Pels with high specific degumming activity and good stability in the conditions used for industrial degumming processes is still required.

In addition to the search for novel enzymes from nature, enzyme engineering, including randomized or irrational (directed evolution or random and combinatorial engineering) and rational (structure based) approaches, has been used to improve the activity and stability of enzymes, including Pels, with varying degrees of success [2,6,18,19]. Relative to directed evolution or random engineering, which rely on high-quality random mutation library construction and high-throughput screening methods, rational design mutation based on structural information and sequence features can be more direct and productive. Several rational strategies based on three-dimensional (3D) structures have been successfully used to predict and improve enzyme thermostability [20,21,22,23]. For example, analysis of B-factors (the B-Fit Method), high-temperature unfolding molecular dynamics simulations, hydrophobic packing improvement of the protein core, and stabilization using disulfide bridges have been used in enzyme stability engineering [20,24,25].

In a previous study, we cloned, overexpressed, and characterized a novel thermostable and alkaline Pel belonging to family PL1 from *Bacillus clausii* S10 (BacPelA) [26]. This enzyme exhibited maximal activity at 70 °C and pH 10.5 and showed high cleavage activity toward methylated pectins. In ramie degumming, significant ramie fiber weight loss was observed following treatment with BacPelA (24.8%) and a combined enzyme-chemical method (30.9%) after a 4 h process, which was the highest degumming efficiency for reported alkaline and thermostable Pels. These characteristics make BacPelA a candidate for industrial use in ramie degumming. However, BacPelA is not very stable in degumming conditions (60 °C and pH 10).

In this work, we determined a high-resolution 3D structure of BacPelA. On the basis of the structure, the B-Fit Method and Δ*T_m_* prediction were used to select mutation sites for the stability improvement of BacPelA. The mutations that were made also improved the activity of the enzyme. We anticipate that our findings will provide a strong foundation for the rational design of alkaline pectate lyase functional characteristics for industrial applications.

## 2. Results and Discussion

### 2.1. Overall Structure of BacPelA

The sequence analysis showed that BacPelA has an N-terminal signal peptide of 23 amino acids. Recombinant mature BacPelA with an N-terminal His-tag was expressed and crystallized in the orthorhombic space group P43212. The crystal structure was solved and refined to 1.78 Å by using the method of molecular replacement with pectate lyase Bsp165PelA from *Bacillus* sp. N16-5 (PDB ID code 3VMV) as the template. Detailed data collection and refinement statistics are presented in Table 1. Almost all the amino acids of BacPelA were observed clearly in electron density maps. The predominant structural motif of BacPelA is a right-handed parallel β-helix formed by three parallel β-sheets (Figure 1). These parallel β-sheets are referred to as PB1, PB2, and PB3, respectively. In total, PB1 has 7 strands, while PB2 and PB3 have 10 and 9 strands, respectively (Figure 1). The loops or turns between the parallel β-sheets, which lack significant secondary structures, are named T1 (connecting PB1 and PB2), T2 (connecting PB2 and PB3), and T3 (connecting PB3 and PB1). T3 is the longest loop extending from the β-sheets which are similar to Bsp165PelA.

Approximately 70 crystal structures of Pels are available in the Protein Data Bank (PDB). They typically have one of two core structure folds, a right-handed β-helix, or an (α/α)_n_ toroid [27]. Although the overall tertiary structure of BacPelA was similar to those of other Pels in PL1, which have the right-handed β-helix fold [4,5,28,29,30,31], there were some differences in the details. A structure complex model with substrate and Ca^2+^ was constructed by superimposing the structure of BacPelA and two highly homologous Pels: Bsp165PelA (PDB: 3VMW) containing pectate trisaccharide substrate (Figure 2A) or EcPelC (PDB: 2EWE) containing pectate tetragalacturonate substrate (Figure 2B). From the superimposed structures, there are two more loops in the T1 and T3 regions of Bsp165PelA (Figure 2A), and one more loop in the T1 region of EcPelC (Figure 2B) than in BacPelA. BacPelA has two more loops in the T3 and *N*-terminal regions than EcPelC. As reported previously, there is little similarity in the detailed structures of the T3 and T1 loops in PL family 1 members [32], while the most obvious difference lies in the long loop extending from the core structure in the T3 region [4,32]. In this region, BacPelA adopts a simple loop similar to that in Bsp165PelA, while EcPelC adopts a compound loop (according to the established taxonomy of loops) [5], and there are three α-helices in the T3 region of EcPelC, while no α-helix was found in this region of BacPelA or Bsp165PelA. In addition, there are three α-helices consisting of more than one turn in the structure of BacPelA, two of which are located on the N- and C-terminal long loops folding along one side of the parallel β-helices, while the shortest is located in the T3 region. In Bsp165PelA and EcPelC, there are four and seven α-helices, respectively.

### 2.2. The Catalytic and Ca^2+^ Binding Sites

All Pels have been reported to share a similar enzymatic mechanism (anti β-elimination), and the essential catalytic base and two supporting basic residues (Lys-Arg-Arg) are conserved in PL 1 pectate lyase [27,33,34]. Sequence alignment of BacPelA (without the signal peptide) with other crystallized PL1 pectate lyases showing high similarity is shown in Figure 3. From the sequence alignment, residues K167, R196, and R201 of BacPelA are conserved and are suggested to be the essential catalytic bases. In the superimposed structures of substrate-binding clefts (Figure 2C,D), the three suggested catalytic residues of BacPelA were highly consistent in position with those of Bsp165PelA (K177, R207, and R212) and EcPelC (K190, R218, and R223). Furthermore, three site-directed mutants of BacPelA were obtained (R196A, R201A, and K167A), and the activity was almost completely lost (data not shown). Hence, amino acids R196, R201, and K167 were confirmed to be the catalytic residues of BacPelA.

In addition to the catalytic residues, some other residues interact with the substrate. R234 and Y269 in Bsp165PelA, corresponding to R245 and Y268 in EcPelC, interact with the substrate by hydrogen bond and stacking interactions, respectively [5,30]. R223 in BacPelA corresponds to R234 of Bsp165PelA and R245 of EcPelC in spatial position and interacts with the substrate via hydrogen bonds. However, Y269 in BspPelA/Y268 in EcPelA is replaced by D269 in BacPelA (Figure 2C,D), which means that this site in BacPelA could not form a stacking interaction with the substrate like that observed in BspPelA and EcPelA. Furthermore, H183 in Bsp165PelA was found to interact with GalpA_-1 [5], but in BacPelA and EcPelC, Ser replaces the His residue at this position (Figure 3), which does not interact with GalpA_-1.

Ca^2+^ is believed to be required for the pectolytic activity of all Pels [24]. Previous reports have identified conserved residues that bind to Ca^2+^ ions or oligogalacturonate in Pel structures [5,28,30,35]. There are two classes of Ca^2+^ in Pel structures: primary Ca^2+^ which binds to the enzyme in the absence of substrate, and additional Ca^2+^, consisting of two or three Ca^2+^ ions, which bridge the enzyme and oligogalacturonate in the complex [30,34,35]. From the sequence alignment (Figure 3), there are three conserved aspartate residues, D113, D143, and D147, for Ca^2+^ binding in BacPelA, which was confirmed by the crystal structure (Figure 1). However, on the basis of the crystal structure, BacPelA likely binds only one Ca^2+^ ion (the primary Ca^2+^), similar to Bsp165PelA, but different from EcPelC, which binds three additional Ca^2+^ ions in the presence of substrate. Although the locations and coordination numbers of the primary Ca^2+^ in the reported structures are the same, there are small differences in the binding sites. In EcPelC, the primary Ca^2+^ coordinates to residues D129, D131, E166, and D170 [30], while in BsPel it coordinates to D183, D222, and D226, similar to Bsp47Pel and BacPelA [28]. In Bsp165PelA, D153 and D157 were found to coordinate the primary Ca^2+^, but the Asp, which is highly conserved in the other Pels and coordinates primary Ca^2+^, is substituted by T121. T121 cannot form an electrostatic interaction with the Ca^2+^ [5].

Considering the additional Ca^2+^ ions, in EcPelC, D160, and D162 participate in a loop interaction with ^2^Ca^2+^ and ^3^Ca^2+^ bridge substrate to both carboxyl oxygens of E166 [30]. E166 is replaced in the corresponding position by D143 in BacPelA (Figure 2D). In addition, in BacPelA, no structurally equivalent residues were found in the loop for ^2^Ca^2+^ binding. Therefore, we suspect that BacPelA does not bind additional Ca^2+^ at these two positions. Although the mutant R218K of EcPelC displayed ^1^Ca^2+^ binding to K218, ^1^Ca^2+^ cannot coordinate with any amino acid in the wild type (it binds only to the substrate). Thus, we considered that ^1^Ca^2+^ might be unimportant for catalysis [5]. Taken together, BacPelA likely binds only one Ca^2+^ ion (the primary Ca^2+^), which might be the reason that a low concentration of Ca^2+^ (0.1 mM) could induce the maximal catalytic activity of BacPelA. This low Ca^2+^ concentration dependence makes BacPelA more attractive for industrial applications.

### 2.3. Construction and Screening of Site-Directed Saturation Mutagenesis Libraries

In a previous study, BacPelA showed high potential application in ramie degumming under alkaline conditions [26]. However, it is not very stable under degumming conditions, which substantially lowers its applicability in the industrial degumming process and must be improved. Although successful approaches for improving the thermostability of pectin hydrolysis enzymes have been reported [2,6,36,37,38], improvement of thermostability by protein engineering remains challenging. However, rational strategies have been successfully applied for the stability improvement of enzymes based on structural information [39,40]. Among them, analysis of B-factors and Δ*T_m_* values are effective for rationally choosing hot-spot residues to mutate to increase enzyme thermostability. The B-factor can be used to identify the flexibility of atoms, side-chains, or even whole regions of proteins, and to rationally identify appropriate (i.e., the most flexible) residues to increase thermostability [41]. Successful rational engineering of thermostability using B-factor-based mutational design has been applied to lipase, pullulanase, xylanase, and sucrose isomerase [42,43,44,45]. Meanwhile, HoTMuSic, a tool that predicts the impact of point mutations on the protein melting temperature (Δ*T_m_*) using the experimental or modeled protein structure as the sole input, was recently applied in thermal stability engineering projects aimed at designing new proteins that feature increased heat resistance or remain active and stable in nonphysiological conditions [46,47]. This tool has been used successfully for thermal stability improvement of enzymes, including lipase, agarase, and arginine deiminase [47,48,49].

On the basis of the structure, the B-factor of each residue of BacPelA was thus valued by B-FITTER, and the sum of positive Δ*T_m_* values per residue was predicted by the HoTMuSiC algorithm (http://dezyme.com/ accessed on 11 September 2019). Six residues displaying the highest average B-factors and four residues with the highest sum of positive Δ*T_m_* values, which were located in unstable regions (loops or turns), were selected for site-directed saturation mutagenesis (SSM) library construction (Table 2). In total, 10 SSM libraries, each containing four 96-well microtiter plates, were constructed. The activity ratio of heat-treated to untreated crude enzyme was designated as the residual activity and was used to screen for mutants with improved thermo-alkaline stability. In the screening condition, wild-type BacPelA served as a control and was almost inactivated.

After preliminary screening, a total of 16 potential mutants showing significant (>10%) higher residual enzyme activities than the wild-type were obtained from the SSM libraries: K139 (two mutants), R150 (four mutants), R216 (three mutants), A238 (five mutants) and R259 (two mutants). By sequencing, eight mutants, K139, R150G, R150S, R216H, R216G, A238C, A238V, and R259L, were identified. These eight mutants were then heterologously expressed in *Escherichia coli* BL21 (DE3) and purified for further stability confirmation. After incubation with 0.1 mg mL^−1^ purified enzyme at 65 °C and pH 10.0 for 15 min, the residual activity of these eight mutant enzymes was determined. R150G, R216H, and A238C showed 33.2%, 42.6%, and 20.8% residual activity, respectively, while the other five mutants and the wild-type enzyme were almost inactivated. Thus, these three mutations resulted in significant improvement in the thermostability of BacPelA in alkaline conditions.

### 2.4. Stability and Enzymatic Properties of the Improved Mutants

The *T_50_^15^* value (the temperature at which the enzyme loses 50% of its initial activity after incubation for 15 min) was used to determine the thermo-alkaline stability of the identified mutants in different conditions. As shown in Table 3, the *T*_50_^15^ value of the purified mutants R150G, R216H, and A238C at pH 10.0 increased by 6.5 °C, 6.0 °C, and 3.0 °C, respectively, compared with the wild-type enzyme. Addition of Ca^2+^ improves the thermo-alkaline stability of wide-type BacPelA. However, similar *T*_50_^15^ value increases were observed for the three mutants in the presence of 0.1 mM Ca^2+^ compared with its absence. The *T*_50_^15^ values of mutants R150G, R216H, and A238C also increased by 2.0 °C, 3.0 °C, and 5.0 °C, respectively, in neutral conditions without the addition of Ca^2+^. The residual activity after treatment in the enzymatic ramie degumming conditions for 4 h (60 °C, pH 10.0, 0.1 mM Ca^2+^) was also assayed (Figure 4). Compared with the wild-type BacPelA, which retained only 3.4% of its original activity, the mutants R150G, R216H, and A238C retained 24.6%, 17.6%, and 33.4% of their original activities, respectively. These results confirmed that these three sites are important for the thermo-alkaline stability of BacPelA.

To elucidate the effects of these three substitutions on enzyme activity, the specific activities and steady-state kinetic parameters of wild-type BacPelA and mutants R150G, R216H, and A238C were determined at 70 °C. As shown in Table 4, the specific activity of R150G, R216H, and A238C was 759.8 U mg^−1^, 833.7 U mg^−1^, and 965.2 U mg^−1^, respectively, increased by 6.7%, 17.1%, and 35.6%, respectively, compared with wild-type BacPelA (711.8 U mg^−1^). Note also that the optimal reaction temperature of mutant R216H was changed to 65 °C, 5 °C lower than that for wild-type BacPelA. At 65 °C, the R216H mutant had a specific activity of 888.3 U mg^−1^, increased by approximately 24.8% compared with wild-type BacPelA at its optimal temperature (70 °C). Therefore, all three mutations not only improved the thermostability of the enzyme but also increased the specific activity.

The specific activity of the mutants was also assayed at the degumming temperature (60 °C) (Table 4). The specific activity of R150G, R216H, and A238C at 60 °C increased by 46.6%, 61.9%, and 24.5%, respectively, compared with the wild-type enzyme. With polygalacturonic acid as the substrate, the *K_m_* values of the R216H and A238C mutants were 0.47 and 0.48 g L^−1^, which were not substantially different from each other or the wild-type enzyme (0.44 g L^−1^). The *K_m_* value of R150G (0.56 g L^−1^) was slightly higher than that of the wild-type enzyme (Table 4). These data indicate that the mutations R216H and A238C did not influence substrate binding, but mutation R150G did slightly. The *k_cat_*/*K_m_* value of mutant R150G decreased by 11.9% compared with the wild-type enzyme. However, the *k_cat_*/*K_m_* values of the other two mutants and the *k_cat_* values of all three mutants significantly increased, to varying degrees, compared with the wild-type enzyme (Table 4). Collectively, these results revealed that the three mutations, R150G, R216H, and A238C, were ideal for improving the thermo-alkaline stability and the catalytic activity of BacPelA.

To our knowledge, this is the first time that substitutions R150G, R216H, and A238C have been identified as contributing to the thermostability and thermo-alkaline stability of Pels. R150 and A238 are conservatively substituted residues in Pels, indicating that these sites might be critical for the thermostability of these enzymes. In addition, the R150G and R216H substitutions resulted in higher Δ*T_50_^15^* values in alkaline conditions than in neutral conditions, which indicated that these residues might also be responsible for the alkaline stability of Pels. R150 is located in the T1 region, and R216 and A238 are located in the T3 region, which are flexible regions in the structure. Therefore, these three substitutions might enhance the rigidity of the protein, thus improving its thermostability. However, decreased conformational flexibility in thermophilic enzymes is generally thought to result in low enzyme activity, while increased flexibility in less stable enzymes is associated with enhanced enzyme activity [2]. Thus, in several cases, thermostability and enzyme activity could not be improved simultaneously [6,50]. However, in the present study, the three point mutations that improved the thermo-alkaline stability of the enzyme also improved the specific activity compared with that of the wild-type enzyme (Table 4). Similar results have been reported in Pels from *Bacillus* sp. N16-5 [2], *B. pumilus* [6], *Xanthomonas campestris* [36], *Paenibacillus* sp. [38], and *Dickeya dadantii* [51], which indicated that the relationship between thermostability and enzyme activity does not have to be opposing.

### 2.5. Combination of the Beneficial Mutations

The stabilizing effects of point mutations could be cumulative [4]. Thus, aiming to achieve greater thermo-alkaline stability and activity of BacPelA, the three beneficial amino acid substitutions were combined with each other. As shown in Table 3, the *T_50_^15^* values of the purified combination mutants R150G/R216H, R150G/A238C, and R216H/A238C at pH 10.0 increased by 11.0 °C, 4.5 °C, 5.5 °C, respectively, without the addition of Ca^2+^, and by 8.5 °C, 6.0 °C, 6.5 °C, respectively, with 0.1 mM Ca^2+^ added, compared with wild-type BacPelA. Furthermore, the *T*_50_^15^ values of these combination mutants increased by 4.0 °C, 6.5 °C, and 6.0 °C, respectively, under neutral conditions. The residual activity of these three combination mutants after treatment in the degumming conditions for 4 h was 50.9%, 75.6%, and 63.1%, 15.0-, 22.2- and 18.6-fold that of the wild-type BacPelA, respectively (Figure 4). Triple-mutant R150G/R216H/A238C had a *T_50_^15^* value of 79.5 °C under neutral conditions, and 67.5 °C and 72.0 °C at pH 10.0 without or with 0.1 mM Ca^2+^, which was 8.0 °C, 8.5°C, and 8.5 °C higher than that of wild-type BacPelA, respectively (Table 4). After treatment in the degumming conditions for 4 h, the mutant displayed a residual activity of 86.2%, which was approximately 25.4-fold that of wild-type BacPelA (Figure 4). The optimal reaction temperature for each of the combination mutants was also determined. The optimal temperature did not change for mutants R150G/A238C and R216H/A238C, while the optimal temperature of the other two combination mutants decreased to 65 °C.

The specific activities and kinetic parameters of these four combination mutants were also determined (Table 4). The specific activities of mutants R150G/A238C, R216H/A238C, and R150G/R216H/A238C at 70 °C were 936.2 U mg^−1^, 935.1 U mg^−1^, and 890.6 U mg^−1^, increased by approximately 31.5%, 31.4%, and 25.1%, respectively, compared with wild-type BacPelA. The activity of the R150G/R216H mutant at 70 °C decreased by around 2.1% (696.9 U mg^−1^) compared with that of the wild type. Moreover, at their optimum reaction temperature of 65 °C, the mutants R150G/R216H and R150G/R216H/A238C showed activities of 854.9 U mg^−1^ and 925.3 U mg^−1^, respectively, which were 20.1% and 30.0% higher than that of wild-type BacPelA at its optimal reaction temperature of 70 °C. At the degumming temperature (60 °C), the specific activity of R150G/R216H, R150G/A238C, R216H/A238C and R150G/R216H/A238C mutants increased by approximately 34.4%, 75.8%, 37.7%, and 86.1%, respectively, compared with that of wild-type BacPelA (Table 4). Therefore, in addition to the thermo-alkaline stability, the specific activities of the combination mutants were significantly improved. The *K_m_* values of R150G/R216H and R150G/A238C mutants were 0.53 and 0.58 g L^−1^, slightly higher than that of wild-type BacPelA, while those of R216H/A238C and R150G/R216H/A238C mutants were 0.34 and 0.31 g L^−1^, lower than that of the wild-type enzyme (Table 4), indicating that these combined mutations influence the substrate binding. Relative to the wild-type, the *k_cat_*/*K_m_* value of all four combined mutants increased to varying degrees (Table 4), especially that of the R150G/R216H/A238C mutant, which more than doubled. Taken together, the triple-mutant R150G/R216H/A238C displayed high potential for ramie fiber degumming because of its high activity and thermo-alkaline stability.

### 2.6. Degumming of Ramie Fibers

A comparative study was performed to evaluate the ramie degumming efficiencies of purified wild-type BacPelA and the mutant R150G/R216H/A238C (hereafter referred to as FCM). Each enzyme was used to degum ramie fibers at 60 °C and pH 10.0 for 4 h using the modified method reported previously [26]. The weight losses of ramie fibers treated with wild-type BacPelA and FCM were 20.5% and 26.7%, respectively (30.2% higher for FCM than for wild-type BacPelA) (Figure 5A). Only approximately 9.5% weight loss was observed in the negative control, in which the ramie fibers were treated only with alkaline buffer. Furthermore, the weight loss on treatment with wild-type BacPelA or FCM was higher than that in the positive control (19.4%), in which ramie fibers were treated with 0.5% (*w*/*v*) NaOH solution. Morphologically, the ramie fibers degummed by FCM were whiter, softer, and more dispersed than the fibers degummed by the wild-type enzyme or NaOH only (Figure 5B). Note that the amount of the FCM used in the ramie degumming experiments in this study was actually lower than that of the wild-type enzyme because of the increased specific activity and thermo-alkaline stability of the FCM under degumming conditions. Overall, only around 80.1% of the amount of wild-type enzyme was used in the FCM degumming, and that resulted in the 30.2% increase in the weight loss of ramie fibers, indicating that BacPelA had been successfully engineered for more efficient ramie degumming by rational engineering.

The enzymatic degumming activity of some other alkaline and thermostable Pels was reported previously. The polygalacturonase from *Bacillus* sp. MG-cp-2 [10] and pectate lyase from *Amycolata* sp. [52] and *B. pumilus* DKS1 [53] showed approximately 20%, 16.7%, and 17% weight loss of ramie fibers after 12, 15, and 24 h of degumming, respectively. Approximately 13.5%, 21.6%, and 23.1% weight loss was found using pectate lyase from *B. subtilis* 7-3-3 [54], *B. pumilus* ATCC7061 [6], and the improved mutant EA from *Bacillus* sp. N16-5 [2] after 4 h of degumming, respectively. Approximately 18.8% weight loss was observed for mutant pectate lyase from *D. dadantii* after 2 h of degumming [51]. Therefore, the degumming efficiency of the mutant FCM of BacPelA (26.9% weight loss after 4 h of treatment) was among the highest reported for alkaline and thermostable Pels. Furthermore, the weight loss of ramie fiber degummed by mutant FCM was significantly higher than that on treatment with 0.5% (*w*/*v*) NaOH solution, indicating its potential application in the substitution of chemical regents. However, the total gum content of the ramie fibers used in this experiment was around 34%, and only fibers with less than 2.5% residual gum can be directly used in further textile processes, and thus a weight loss of more than 30% is required [1,55]. In previous research, BacPelA-chemical combined degumming increased the weight loss of ramie fibers to 30.9%, and decreased the use of NaOH by approximately 60% compared with the traditional chemical degumming process [26]. Our findings indicate that the mutant FCM enzyme could further improve degumming efficiency and reduce the requirement for chemical reagents (e.g., NaOH) in a combined enzyme-chemical process, confirming its potential for application in the textile industry.

## 3. Materials and Methods

### 3.1. Strains, Plasmids, and Chemicals

*E. coli* BL21 (DE3) (TransGen Biotech, Beijing, China) and pET28a (Novagen, Temecula, CA, USA) were used as the host strain and vector for gene expression, respectively. Polygalacturonic acid (PGA) was purchased from Sigma-Aldrich (St. Louis, MO, USA). All enzymes for DNA manipulation were purchased from New England Biolabs (Ipswich, MA, USA). Isopropyl-β-D-thiogalactopyranoside (IPTG), kanamycin, and imidazole were from Amresco Inc. (Solon, OH, USA). All other chemicals were of reagent grade.

### 3.2. Protein Expression, Purification, and Enzyme Assay

Recombinant BacPelA and mutants with *N*-terminal His-tag were expressed and purified as described previously [26], except that 10 mM Tris-HCl buffer (pH 7.5) was used as the desalting buffer. Purity was assessed by sodium dodecyl sulfate polyacrylamide gel electrophoresis (SDS-PAGE) with 10% separating gel. The protein concentration was determined using a Quick Start Bradford protein assay kit (Bio-Rad, Hercules, CA, USA), with bovine serum albumin (0.125–1.0 mg mL^−1^) as standard. Recombinant BacPelA was concentrated to 20 mg mL^−1^ for crystallization.

The enzyme activity of purified BacPelA and mutants was determined by measuring the increase in produced unsaturated bonds at 235 nm (A_235_ method). The reaction mixture consisted of 190 μL of 50 mM glycine–NaOH buffer (pH 10.5) containing 0.2% PGA (*w*/*v*) and 0.1 mM CaCl_2_ and 10 μL of appropriately diluted enzyme. The reaction was performed at 70 °C for 10 min and stopped by adding 300 μL of 30.0 mM H_3_PO_4_. Absorbance at 235 nm was measured with a Spectra Max 190 Microplate Reader (Molecular Devices, Sunnyvale, CA, USA). One unit of Pel activity was defined as the amount of enzyme producing 1 μmol of unsaturated bonds in 1 min with a molar extinction coefficient of 4600.0 M^−1^·cm^−1^. All activity measurements were repeated three times.

### 3.3. Crystallization, Data Collection, and Structure Analysis

Concentrated BacPelA was subjected to crystallization screening using the sitting-drop vapor-diffusion method from Hampton Research (Laguna Niguel, CA, USA) at 25 °C. In general, 1 μL of 6.5 mg mL^−1^ BacPelA was mixed with 1 μL of reservoir solution in 48-well Cryschem Plates, and then equilibrated against 100 μL of the reservoir at 25 °C. The crystals of BacPelA were obtained under the following conditions: 0.05 M MgCl_2_, 0.1 M HEPES pH7.5, 30% PEG550, within 2 to 5 days, the crystals reached dimensions suitable for X-ray diffraction. All of the X-ray diffraction data were tested and collected at beam lines BL10U2, BL17B, BL18U1, and BL19U1 of the National Facility for Protein Science in Shanghai (NFPS) at the Shanghai Synchrotron Radiation Facility (SSRF). The crystals were mounted in a cryoloop, soaked with cryoprotectant solution (0.05 M MgCl_2_, 0.1 M HEPEs pH7.5, 30% PEG550, 10% glycerol) prior to data collection at 100 K. The diffraction images were processed using HKL2000 [56]. All the crystal structures were solved by the molecular replacement (MR) method with the Phaser program [57] from the Phenix suite [58] using the structure of pectate lyase Bsp165PelA from *Bacillus* sp. N16-5 as the search model (PDB code 3VMW) [5]. Further refinement was carried out using programs phenix.refine [59] and Coot [60]. Prior to structural refinements, 5% randomly selected reflections were set aside for calculating Rfree as a monitor. All figures were prepared using the PyMOL [61].

### 3.4. Site-Directed Saturation Mutagenesis (SSM) and Screening

SSM was performed using a previously published protocol [62]. Degenerate primers containing appropriate base substitutions were used for saturation mutagenesis. PCR was carried out with a 50 ng template plasmid (pET28a-*pelA*), 1 μmol primer pairs, 0.2 mM dNTPs, and 1 U Pyrobest DNA polymerase in a final volume of 50 μL. The following PCR profile was used: 1 cycle of 94 °C for 4 min; 20 cycles of 94 °C for 45 s, 55 °C for 1 min, and 72 °C for 7 min; followed by a final extension at 72 °C for 20 min. The PCR products were then digested by *DpnI* overnight. The products were transformed into *E. coli* BL21 (DE3) competent cells by electroporation transformation and plated onto LB-kanamycin (60 μg mL^−1^) agar plates for further screening. Single colonies from the plates were picked with sterilized toothpicks, plated in 160 μL LB medium containing kanamycin (50 μg mL^−1^) in sterilized 96-well microplates, and incubated overnight at 37 °C with shaking at 900 rpm. Fifty microliters of sterilized 60% glycerol were then added to each well, and the plates were stored at −70 °C to create the SSM libraries.

Each mutant in the microplate was replicated in 200 μL LB medium containing kanamycin (60 μg mL^−1^) and IPTG (0.1 mM) prepared in a microplate using a sterile 96-pin replicator. After cultivation for 15 h at 37 °C, 10 μL of RIPA lysis buffer was added to each well, and the mixture was incubated at 37 °C for 30 min. Then 10 μL of lysate was added to 90 μL of 50 mM glycine–NaOH buffer (pH 10.0) in 96-well PCR plates, and heat-treated at 80 °C for 15 min. After incubation, residual enzyme activities were assayed immediately using the 3,5-dinitrosalicylic acid (DNS) method in new 96-well PCR plates. The reaction mixture in each well contained 70 μL of 50 mM glycine–NaOH buffer (pH 10.5), 0.2% PGA (*w*/*v*), and 0.1 mM CaCl_2_, and 10 μL of heat-treated variant enzyme. The reaction was performed at 70 °C for 10 min and stopped by adding 80 μL of DNS solution, followed by incubation at 98 °C for 8 min. Variants showing a significant increase in residual enzyme activities compared with the control (i.e., the wild type) were regarded as candidate mutants with improved thermo-alkaline stability.

### 3.5. Determination of Thermostability and Kinetic Parameters

The *T*_50_^15^ value, which is the temperature at which the enzyme has lost 50% of its initial activity after 15 min incubation, was determined to evaluate the thermo-alkaline stability of the wild-type and mutated enzymes. Purified enzymes were adjusted to a concentration of about 0.1 mg mL^−1^ and then incubated in 50 mM glycine–NaOH buffer (pH 10.0) with or without Ca^2+^ at different temperatures for 15 min. The residual activity was assayed using the A_235_ method under standard conditions.

The kinetic parameters of wild-type and mutant BacPelA were determined as previously described [26] using the A_235_ method at the optimal reaction temperature and pH. The substrate concentrations of PGA were 0.1–3.0 mg mL^−1^ for both wild-type BacPelA and mutant enzymes. All data are the averages of triplicate measurements. Kinetic parameters were calculated using GraphPad Prism 5.0 software and non-linear regression. All data are expressed as the means of triplicate measurements.

### 3.6. Enzymatic Degumming of Ramie Fibers

The degumming of ramie fibers by wild-type BacPelA and its mutants was evaluated by measuring the weight loss of ramie fibers. The experiments were performed using modified Zhou’s method [26]. Approximately 20.0 U mL^−1^ of purified wild-type or mutant enzyme was used to degum 1.5 g of ramie at 60 °C for 4 h in 30 mL of 50 mM glycine–NaOH buffer (pH 10.0) containing 0.1 mM CaCl_2_. Ramie fibers treated only with alkaline buffer (pH 10.0) or 0.5% (*w*/*v*) NaOH solution were used as negative and positive control, respectively. After enzymatic treatment, the fibers were washed twice with water after beating to remove residual gum from the surface of the ramie fibers and then dried to a constant weight at 105 °C. All experiments were repeated three times.

## 4. Conclusions

In this study, the crystal structure of the alkaline pectate lyase BacPelA from alkaliphilic *B. clausii* S10 was solved at a high resolution (1.78 Å). The overall structure was similar to that of other members of the PL1 family, but there were some differences in the details of the secondary structure and Ca^2+^ binding sites. The structure suggests that BacPelA binds only to the primary Ca^2+^, which may explain why this enzyme shows low dependence on the Ca^2+^ concentration. This characteristic is attractive for industrial applications. On the basis of the crystal structure, BacPelA was engineered by applying B-factor and positive Δ*T_m_* analysis and SSM, aiming to improve the thermo-alkaline stability. Three key single substitutions, R150G, R216H, and A238C, were identified as improving the thermostability and thermo-alkaline stability of BacPelA. The mutant in which these three point mutations were combined, named FCM, exhibited a Δ*T*_50_^15^ value of 8.5 °C compared with that of the wild-type enzyme in thermo-alkaline conditions, and a 25.1% increase in catalytic activity. FCM was also more efficient than wild-type BacPelA in ramie degumming. Thus, our research has produced a thermo-alkaline stable and highly active pectate lyase with great potential for application in the textile industry. It also illustrates a rational design strategy for improving the thermo-alkaline stability and activity of pectate lyases. However, the molecular mechanism for thermo-alkaline stability and activity improvement of these key single substitutions is not very clear yet. So the molecular dynamics (MD) simulation and structural analysis need to be done in the future to reveal the mechanism. Moreover, the FCM-chemical degumming needs to be further evaluated in the ramie fiber degumming process.

## Figures and Tables

**Figure 1 ijms-24-00538-f001:**
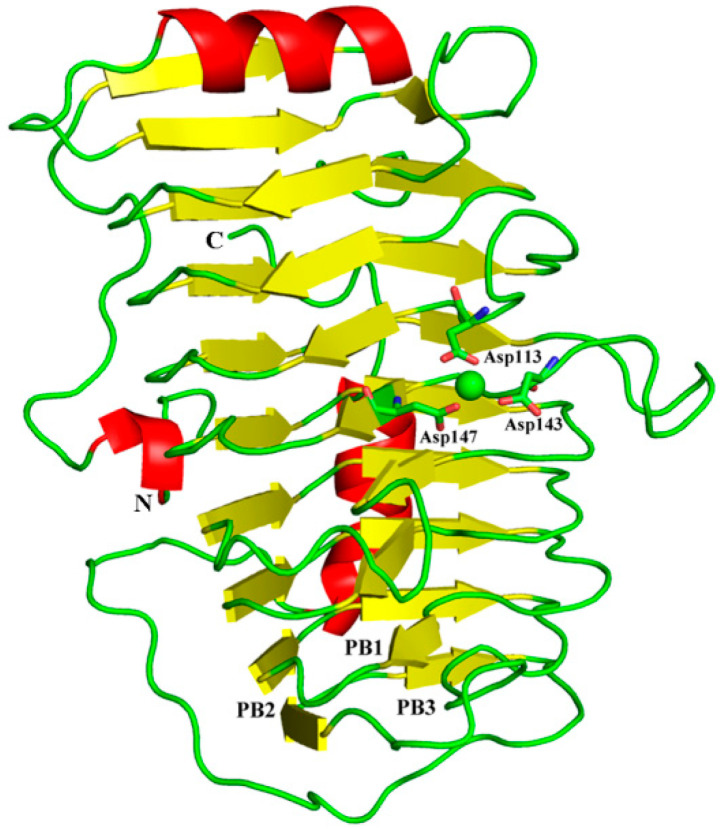
Overall structure of BacPelA. The three β–sheets are denoted as PB1, PB2, and PB3. Ca^2+^ ion is shown as a green sphere.

**Figure 2 ijms-24-00538-f002:**
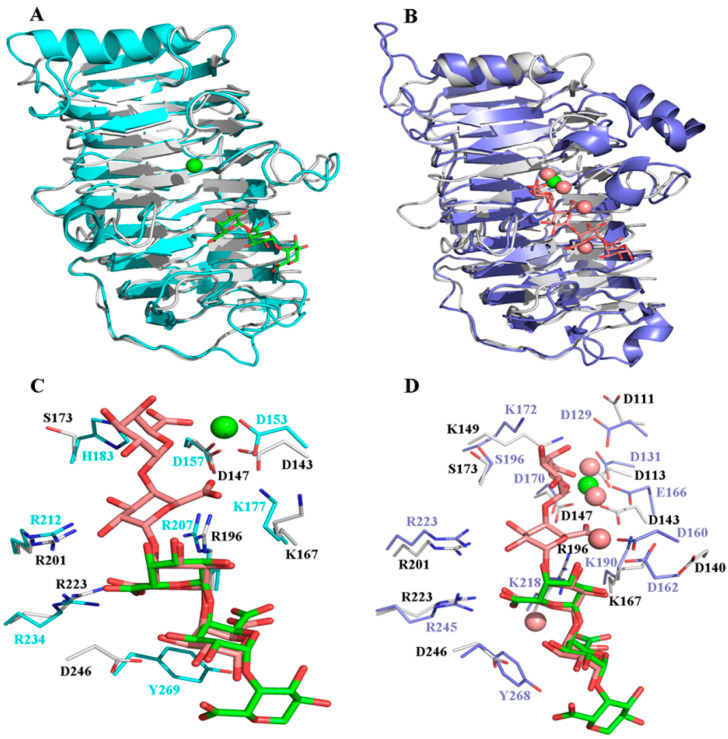
Structural comparison of BacPelA with BspPelA-TGA-Ca^2+^ and EcPelC-tetragalacturonate-Ca^2+^. (**A**) The BacPelA (gray) is superimposed with BspPelA-TGA-Ca^2+^ (blue; PDB ID: 3VMW). (**B**) The BacPelA (gray) is superimposed with EcPelC-tetragalacturonate-Ca^2+^ (purple; PDB ID: 2EWE). (**C**) Detailed interaction comparison in the substrate and Ca^2+^ binding clefts of BacPelA and BspPelA. (**D**) Detailed interaction comparison in the substrate and Ca^2+^ binding clefts of BacPelA and EcPelC. TGA and tetragalacturonate (TetraGal*p*A) are shown in green and brown, respectively. The only Ca^2+^ ion and the four Ca^2+^ ions are shown as green sphere and brown spheres, respectively.

**Figure 3 ijms-24-00538-f003:**
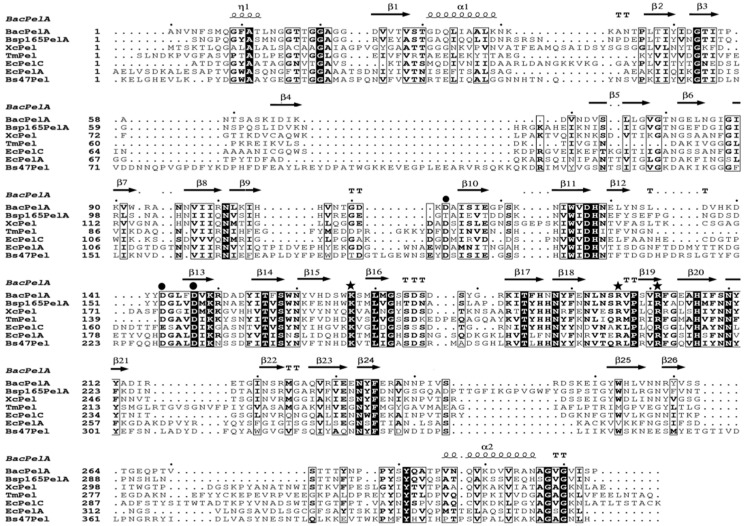
Multiple amino acid sequence alignment of crystalized PL1 pectate lyase. The pectate lyases used were BacPelA, Bsp165PelA from *Bacillus* sp. 16-5 (PDB 3VMV), XcPel from *Xanthomonas campestris* (PDB 2QX3), TmPel from *Thermotoga maritima* (PDB 3ZSC), EcPelA and EcPelC from *Erwinia chrysanthemi* (PDB 1PE9 and 2EWE), and Bs47Pel from *Bacillus* sp. TS-47 (PDB 1VBL). Strictly conserved residues are shaded black, and conservatively substituted residues are boxed. The secondary structural elements (helices-α, strands-β, turns-T, and helices-η) of BacPelA are shown above the aligned sequences. The conserved catalytic sites and calcium binding sites are indicated by asterisks and circles, respectively. The figure was produced using ESPript 3.0 (http://espript.ibcp.fr/ESPript/ESPript/index.php, accessed on the 20 April 2022).

**Figure 4 ijms-24-00538-f004:**
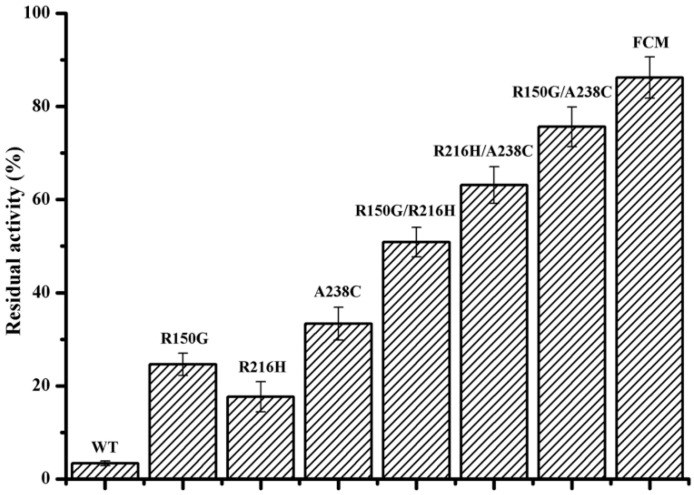
The residual activity of wild-type BacPelA and mutant enzymes under degumming conditions. The residual activity was determined after incubation at 60 °C and pH 10.0 with 0.1 mM Ca^2+^ for 4 h. FCM means the final combined mutant R150G/R216H/A238C. The measurements were performed in three independent experiments. Error bars represent standard deviations.

**Figure 5 ijms-24-00538-f005:**
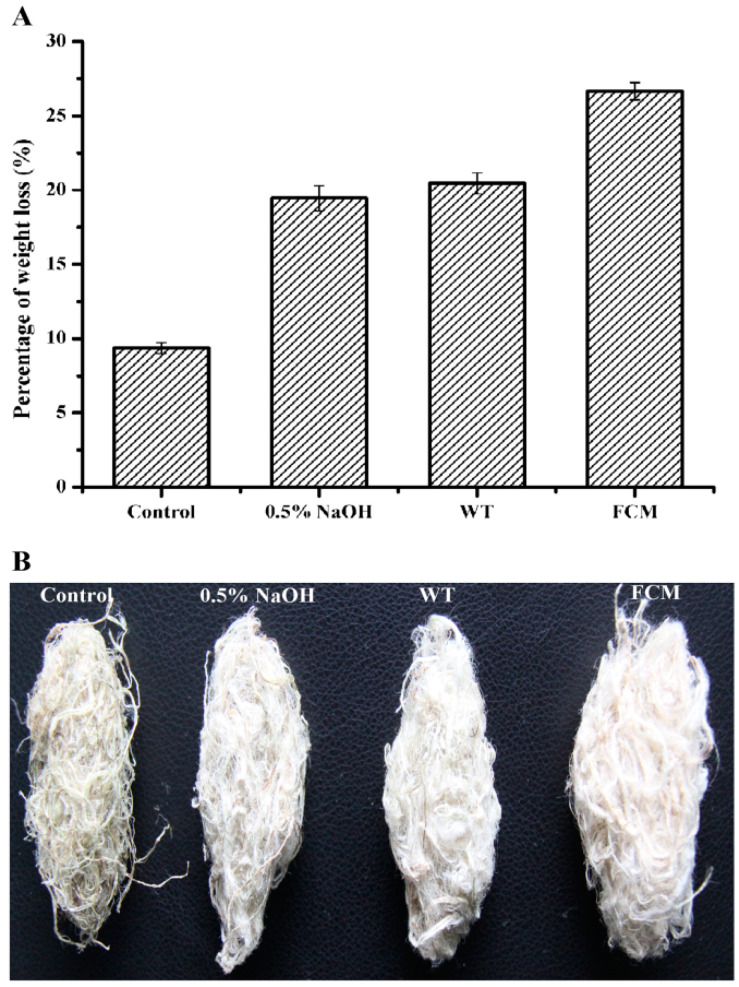
Percent weight loss and morphological image of degummed ramie fibers. (**A**) The ramie fiber weight loss. (**B**) The morphological image of degummed ramie fibers. FCM means the final combined mutant R150G/R216H/A238C. The weight loss measurements were performed in three independent experiments. Error bars represent standard deviations.

**Table 1 ijms-24-00538-t001:** Summary of X-ray data collections and refinement statistics.

Parameter	BacPelA
*Data collection*	
Space group	*P43212*
*Unit-cell parameters*	
*a*, *b*, *c* [Å]	109.08, 109.08, 44.97
*α*/*β*/*γ* (°)	90.00/90.00/90.00
Resolution (Å)	30–1.78 (1.84–1.78)
Unique reflections	26611 (2609)
Redundancy	10.4 (10.2)
Completeness (%)	100 (100)
Average *I*/*σ*(*I*)	39.1 (3.57)
CC 1/2	0.99 (0.84)
*Refinement*	
R_work_ (95% data)	0.154 (0.189)
R_free_ (5% data)	0.189 (0.237)
Rmsd bonds (Å)	0.010
Rmsd angles (°)	1.12
*Dihedral angles*	
Most favored (%)	93.9
Allowed (%)	6.1
Disallowed (%)	0.00
*Number of non-hydrogen atoms/Average B-factor*	
Protein	2353/21.48
Ion/Ligands	43/37.69
solvent	179/33.29
PDB ID code	7XKS

**Table 2 ijms-24-00538-t002:** Mutation sites and primers used for PCRs.

Site (Average B-Factors)	Location	Primers	Sequence (5′-3′)
Lys139 (51.37)	T3	Forward	AGCCTTGATGTGCATNNKGACTATTATGATGGATTGTTTG
		Reverse	TCCATCATAATAGTCMNNATGCACATCAAGGCTATTATAG
Gln282 (44.82)	C-terminal long loop	Forward	TCCACCATACAGCTATNNKGCAACGCCTGTAAACCAAGTG
		Reverse	GTTTACAGGCGTTGCMNNATAGCTGTATGGTGGATTATAAG
Arg259 (41.39)	T2	Forward	CATCTGGTCAATAATNNKTATGTTTCTTCGACTGGCG
		Reverse	CCAGTCGAAGAAACATAMNNATTATTGACCAGATGCC
Glu266 (38.47)	C-terminal long loop	Forward	ATGTTTCTTCGACTGGCNNKCAGCCAACTGTCTCGAAC
		Reverse	TCGAGACAGTTGGCTGMNNGCCAGTCGAAGAAACATAG
Arg150 (37.82)	T1	Forward	GATTGTTTGATGTGAAANNKGATGCTGATTACATTAC
		Reverse	TAATGTAATCAGCATCMNNTTTCACATCAAACAATCC
Arg216 (35.70)	T3	Forward	ATTACTACGCAGATATTNNKGAGACAGGGATCAATTCTC
		Reverse	GAATTGATCCCTGTCTCMNNAATATCTGCGTAGTAATTG
**Site (sum of positive Δ*T_m_*)**	**Location**	**Primers**	**Sequence (5** **′** **-3** **′** **)**
Ser280 (18.57)	C-terminal long loop	Forward	GACTTATAATCCACCATACNNKTATCAAGCAACGCCTG
		Reverse	TACAGGCGTTGCTTGATAMNNGTATGGTGGATTATAAG
Ala238 (13.84)	T3	Forward	AAAACTATTTTGAAAGGNNKAACAATCCAATTGTAAGC
		Reverse	CTTACAATTGGATTGTTMNNCCTTTCAAAATAGTTTTC
Gly23 (12.06)	N-terminal long loop	Forward	TACAGGGGGTGCTGGANNKGATGTTGTAACCGTTTCTAC
		Reverse	GAAACGGTTACAACATCMNNTCCAGCACCCCCTGTAGTC
Ala94 (10.42)	T1	Forward	TGGCATTAAAGTATGGCGGNNKAATAACGTGATCATCCG
		Reverse	TGCGGATGATCACGTTATTMNNCCGCCATACTTTAATGC

**Table 3 ijms-24-00538-t003:** Optimal temperature and *T*_50_^15^ value of the wild-type BacPelA and mutants.

Mutant	Optimal Temp (°C)	*T*_50_^15^ Value (pH 7.5) (°C)	*T*_50_^15^ Value (pH 10) (°C)	*T*_50_^15^ Value (pH 10, 0.1 mM Ca^2+^) (°C)
WT	70	71.5	59.0	63.5
R150G	70	73.5	65.5	70.0
R216H	65	74.5	65.0	70.5
A238C	70	76.5	62.0	66.5
R150G/R216H	65	75.5	70.0	72.0
R150G/A238C	70	78.0	63.5	69.5
R216H/A238C	70	77.5	64.5	70.0
R150G/R216H/A238C	65	79.5	67.5	72.0

**Table 4 ijms-24-00538-t004:** Activity and kinetic parameters of the wild-type BacPelA and mutants.

Mutant	Specific Activity at 70 °C (U mg^−1^)	Specific Activity at 60 °C (U mg^−1^)	*K_m_* (g L^−1^)	*k_cat_* (s^−1^)	*k_cat_*/*K_m_* (s^−1^ L g^−1^)
WT	711.8	482.4	0.44	192.1	436.6
R150G	759.8	707.2	0.56	215.5	384.8
R216H	833.7	781.0	0.47	249.6	531.1
A238C	965.2	600.6	0.48	299.2	623.3
R150G/R216H	696.9	648.3	0.53	268.3	506.2
R150G/A238C	936.2	848.2	0.58	375.6	647.6
R216H/A238C	935.1	664.3	0.34	287.1	844.4
R150G/R216H/A238C	890.6	897.7	0.31	284.2	916.8

## Data Availability

Not applicable.

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
