# Peer review of "Structure of an Alkaline Pectate Lyase and Rational Engineering with Improved Thermo-Alkaline Stability for Efficient Ramie Degumming"

_ijms, 2022, doi:10.3390/ijms24010538_

Round 1
Reviewer 1 Report
The authors reported that the crystal structure of the alkaline pectate lyase (BacPelA) was solved with high resolution (1.78 Å). The authors also have provided a detailed analysis of the structural features of BacPelA, such as its low Ca2+ dependence characteristic. In addition, the site-directed saturation mutagenesis was conducted for improving the thermo-alkali stability and activity of BacPelA. Overall, BacPelA showed great potential for textile industrial application. The subject of the study is interesting and the text is generally well-written. Therefore, I recommend for publication of the work.
Some specifical comments are mentioned below:
The Materials and Methods Section should be placed before the Results and discussion Section.
Author Response
Dear Sir,
Thank you for your valuable and very helpful comments. We have revised the manuscript and the revised sections were highlighted in yellow in the revised manuscript. The English writing has also been improved by a professional language service.
Specific comments:
- The Materials and Methods Section should be placed before the Results and discussion Section.
Answer: Thanks for your suggestion. Actually, according to the Microsoft Word template and the article format of IJMS, the Materials and Methods section follows the Results and Discussion section. So we do not revise this in the new manuscript.
Reviewer 2 Report
In the manuscript, the authors recovered the structure of one pectate lyases, BacPelA. The authors selected several sites for mutation. Three mutant enzymes showed increase pH and thermal tolerance, and the activity was increased. The authors used the enzyme to treat ramie fibers, and it can remove gum in the ramie fibers. The content is well, and the experiment design is OK. However, I strongly suggest that the editor should give the authors 15 days to edit the language of this manuscript thoroughly. Too many language errors are available in current version of the manuscript. For example
1, Line 17, there should be a comma before such as.
2, Line 18, showing potential for enzymatic, the authors should notice the sentence.
3, Line 24-26, this sentence is not acceptable.
4, Line 34, increase on specific activity?
Moreover, many sentence are too long, and they should be revised to be short and clear.
It seemed the authors did not submit the protein structure to the database, and no public accession number was provided.
Author Response
Dear Sir,
Thank you for your valuable and very helpful comments. We have revised the manuscript throughout to address your comments described in detail below and the revised sections were highlighted in yellow in the revised manuscript. The English writing has also been improved by a professional language service.
Here are our point-by-point responses:
- However, I strongly suggest that the editor should give the authors 15 days to edit the language of this manuscript thoroughly. Too many language errors are available in current version of the manuscript.
Answer: Thanks for your suggestion. The English writing of the manuscript has been improved thoroughly by a professional language service.
- Line 17, there should be a comma before such as.
Answer: Thanks for your questions. We have revised this in the new manuscript.
- Line 18, showing potential for enzymatic, the authors should notice the sentence.
Answer: Thanks a lot for your suggestion. We have revised this sentence in the new manuscript.
- Line 24-26, this sentence is not acceptable.
Answer: Thanks for your suggestion. We have revised this sentence in the new manuscript.
- Line 34, increase on specific activity?
Answer: Actually, here we mean that the ramie fiber weight loss degummed by the combined mutant (26.7%) increased by 30.2% compared with that degummed by the wild-type BacPelA (20.5%). This sentence was also revised in the new manuscript.
- Moreover, many sentence are too long, and they should be revised to be short and clear.
Answer: Thanks for your suggestion. The English writing of the manuscript has been improved thoroughly by a professional language service and some long sentences have been also revised in the new manuscript.
- It seemed the authors did not submit the protein structure to the database, and no public accession number was provided.
Answer: Actually, we submitted the protein structure to the PDB data base and the PDB ID code was 7XKS. This code was mentioned in Table 1 in the manuscript.
Reviewer 3 Report
I suggest authors to address the following issues before this manuscript is considered for publication in the journal.
- Line 50: “Pels… are the major virulence factors in plant pathogens.” Line 55: “Pels have many potential and important industrial applications, such as food production and textile processes”. Authors may explain if Pels virulence activity can affect their industrial applications.
- Authors may define ‘degummimg’ and its importance with respect to this study for general readers.
- Once the genus Bacillus used in the first appearance, it can be abbreviated afterwards (e.g., B. pumilus).
- Did the authors deposit or have plans to do so the results of this study in the PDB or other database?
- Authors may include limitations and/or future works related with this study.
- Citing several literatures published before the last two decades may raise some issues related to the novelty of the work or other related doubts.
Author Response
Dear Sir,
Thank you for your valuable and very helpful comments. We have revised the manuscript throughout to address your comments described in detail below and the revised sections were highlighted in yellow in the revised manuscript. The English writing has also been improved by a professional language service.
Here are our point-by-point responses:
- Line 50: “Pels… are the major virulence factors in plant pathogens.” Line 55: “Pels have many potential and important industrial applications, such as food production and textile processes”. Authors may explain if Pels virulence activity can affect their industrial applications.
Answer: Thanks for your suggestion. Actually, plant cell walls are primarily polysaccharide in composition. A simple but major pathogenic mechanism in plants involves degradation of the cell wall by a battery of polysaccharidases secreted by pathogens. Pels can degrade the pectate network of plant cell walls, so they are considered major virulence factors in plant pathogens. However, industrial application of Pels is to use their catalytic activity on degradation of pectate or pectin. Therefore, the virulence activity (pectate or pectin degradation) does not affect their industrial application.
- Authors may define ‘degumming’ and its importance with respect to this study for general readers.
Answer: Thanks for your good suggestion. We have added some description of degumming in the introduction section in the new manuscript.
- Once the genus Bacillus used in the first appearance, it can be abbreviated afterwards (e.g., B. pumilus).
Answer: Thanks for your suggestion. We have revised these in the new manuscript.
- Did the authors deposit or have plans to do so the results of this study in the PDB or other database?
Answer: Actually, we have submitted the protein structure to the PDB data base and the PDB ID code was 7XKS. This code was mentioned in Table 1 in the manuscript.
- Authors may include limitations and/or future works related with this study.
Answer: Thanks for your suggestion. We have added some content about the limitations and/or future works related with this study in the Conclusion section.
- Citing several literatures published before the last two decades may raise some issues related to the novelty of the work or other related doubts.
Answer: Thanks for your suggestion. The references published before 2000 have been deleted or substituted by new references in the new manuscript.
Round 2
Reviewer 2 Report
If possbile, further language polishment would be better.